# Hypoxia and Hypoxia-Inducible Factor Signaling in Muscular Dystrophies: Cause and Consequences

**DOI:** 10.3390/ijms22137220

**Published:** 2021-07-05

**Authors:** Thuy-Hang Nguyen, Stephanie Conotte, Alexandra Belayew, Anne-Emilie Declèves, Alexandre Legrand, Alexandra Tassin

**Affiliations:** 1Laboratory of Respiratory Physiology, Pathophysiology and Rehabilitation, Research Institute for Health Sciences and Technology, University of Mons, 7000 Mons, Belgium; thuyhang.nguyen@umons.ac.be (T.-H.N.); stephanie.conotte@outlook.com (S.C.); alexandra.belayew@umons.ac.be (A.B.); alexandre.legrand@umons.ac.be (A.L.); 2Department of Metabolic and Molecular Biochemistry, Research Institute for Health Sciences and Technology, University of Mons, 7000 Mons, Belgium; anne-emilie.decleves@umons.ac.be

**Keywords:** hypoxia, myopathies, HIF-1α

## Abstract

Muscular dystrophies (MDs) are a group of inherited degenerative muscle disorders characterized by a progressive skeletal muscle wasting. Respiratory impairments and subsequent hypoxemia are encountered in a significant subgroup of patients in almost all MD forms. In response to hypoxic stress, compensatory mechanisms are activated especially through Hypoxia-Inducible Factor 1 α (HIF-1α). In healthy muscle, hypoxia and HIF-1α activation are known to affect oxidative stress balance and metabolism. Recent evidence has also highlighted HIF-1α as a regulator of myogenesis and satellite cell function. However, the impact of HIF-1α pathway modifications in MDs remains to be investigated. Multifactorial pathological mechanisms could lead to HIF-1α activation in patient skeletal muscles. In addition to the genetic defect *per se*, respiratory failure or blood vessel alterations could modify hypoxia response pathways. Here, we will discuss the current knowledge about the hypoxia response pathway alterations in MDs and address whether such changes could influence MD pathophysiology.

## 1. Introduction

Muscular dystrophies (MDs) are a heterogeneous group of inherited degenerative muscle disorders resulting in progressive muscle weakness and dystrophic histopathology observed on muscle biopsies. Clinically, MDs are characterized by a high variability in terms of age of onset, severity, and progression. MDs are also very heterogeneous in their genetic features and the distribution of the affected muscles, including or not an impact on cardiac or respiratory muscles as well as extra-muscular manifestations such as insulin resistance. 

Respiratory impairments are frequent MD clinical manifestations and associated to hypoxemia in subgroups of patients [1,2,3]. At the tissue level, hypoxemia leads to cellular hypoxia. In primary muscle disorders, the impact of hypoxia on muscle pathophysiology remains poorly documented [4]. This is particularly surprising since several MDs are known to cause chronic hypoxemia due to hypoventilation as a consequence of respiratory muscle weakness.

The Nobel prize in Physiology or Medicine was awarded in 2019 to William G. Kaelin Jr, Peter J. Ratcliffe and Gregg L. Semenza “for their discoveries of how cells sense and adapt to oxygen availability” [5]. They uncovered the main effectors of the hypoxic response, the HIF transcription factor family, and how oxygen-dependent post-translational modifications of HIF-1/2 α lead to their degradation in normoxic conditions and their activation in hypoxia. HIF factors (HIF-1, HIF-2, HIF-3) are heterodimers composed of an oxygen-regulated α subunit and a stable β subunit. HIF-1α and HIF-2α are considered as the master regulators of the hypoxic response transcriptional program. Under normoxic conditions, HIFα subunits are constantly expressed but rapidly degraded by a complex mechanism. They are first hydroxylated by specific Prolyl Hydroxylase Domain-containing enzymes (PHD). This reaction is oxygen-dependent, since the transferred hydroxyl group is derived from the O_2_ molecule. This reaction also requires three cofactors, namely 2-oxoglutarate, vitamin C, and iron [6]. Then, the Von Hippel–Lindau (pVHL) E3 ubiquitin-ligase recognizes hydroxylated HIFα forms and activates their ubiquitination leading to their degradation by the proteasome. By contrast, under hypoxic conditions, PHD cannot hydroxylate HIFα because of the decreased O_2_ availability, allowing for HIFα stabilization, dimerization with HIF1β, and translocation into the nucleus. HIF factors then activate the transcription of more than a hundred target genes through their binding to a specific DNA sequence called Hypoxia Response Element (HRE) [7,8,9,10] (Figure 1 and Table 1). HIF-1α and HIF-2α belong to the basic helix-loop-helix (bHLH) family of transcription factors and exhibit a highly conserved structure and functional similarities. However, an increasing number of studies point to differences in transcriptional regulation and function in response to hypoxia and in disease states [11]. The role of HIF-3 is still debated since its gene can express multiple variants exhibiting different activities, some of which were reported as negative regulators of HIF-1/2α [12]. 

Compared to other oxygen-sensitive tissues such as the brain and heart, skeletal muscles tolerate hypoxia quite well because of their plasticity. Indeed, they adapt their metabolism and structural features (fiber size, muscle fiber type, mitochondrial activity, myoglobin content) as well as blood supply (capillary density) in response to hypoxia, a condition muscles encounter in non-pathological contexts e.g., physical exercise or exposure to high altitude [13,14,15]. Obviously, muscle adaptation will strongly depend on exercise training duration and modalities (resistive vs. endurant). While muscle response to hypoxia has been studied in physiological conditions [16,17], the causes and consequences of hypoxia or HIF-1α pathway activation in the particular context of MDs remain to be clarified. In this review, we provide an overview of the potential causes of hypoxia in skeletal muscle dystrophies. We also address how hypoxia or HIF-1α pathway activation may influence skeletal muscle pathophysiology in MDs and discuss potential avenues for future investigations and therapeutic options.

## 2. Causes of Hypoxia and HIF-1α Pathway Activation in MDs

### 2.1. Respiratory Complications in Muscular Dystrophy

Respiratory failure is a common feature in almost all forms of MDs and the main cause of death in these patients, together with heart dysfunction (see Figure 2 and Table 2). It is defined as the inability to perform adequately the fundamental function of the respiratory system: i.e., to provide proper oxygenation (referred to as oxygenation failure) and carbon dioxide elimination (referred to as ventilatory failure). Negative impact on respiratory functions considerably impairs life quality and adds to the disease burden. While the onset and the degree of respiratory impairment are variable according to the MD form and progression, they are generally described to develop with disease severity and after ambulation loss. It is also important to mention that, even among patients with the same genetic disorder, there is a high variability in the age of onset, severity and progression of the respiratory impairment [1,18,19,20,21,22]. Indeed, in some MDs such as Duchenne Muscular Dystrophy (DMD), Myotonic Dystrophy (DM) or Limb-Girdle Muscular Dystrophy (LGMD), respiratory alterations appear at an early stage of the disease and are a part of the pathological phenotype itself and a major cause of mortality and morbidity. However, in MDs such as Facioscapulohumeral Muscular Dystrophy (FSHD) and Emery–Dreifuss Muscular Dystrophy (EDMD), respiratory troubles are associated with long lasting or severe disease [22].

Respiratory failure in MDs can be classified into two forms mainly determined by the cause of the respiratory impairments: lung failure and failure of the respiratory system pump [21]. 

Basically, respiration is based on the exchange and renewing of gas in the lung. Preventing aspiration of fluid and secretions in the lung and maintaining airway clearance ensure the protection of these physiological processes. Most of the MDs favor these alterations leading to mucus accumulation, obstruction of the bronchi with atelectasis and lower airway infection. Recurrence of pneumonia is then the main cause of pulmonary failure in these diseases [21,22]. Cough is the corner stone of secretion clearance in the lung and upper airways. Deep inspiration, closure of the glottis, high intrathoracic pressure and expiratory flow are needed to allow its efficacy. Presence of an inspiratory muscle weakness and decreased chest wall compliance prevent patients from taking the deep breath required for an effective cough. Moreover, bulbar muscles are critical to keep the glottis closed at the beginning of the expiratory phase (allowing the huge increase in expiratory pressure). Finally, the high expiratory pressure is generated by the strength of expiratory muscle contraction that is frequently reduced in MDs. Inspiratory, expiratory and bulbar muscle weakness in MDs, therefore, prevent efficient cough and lead to secretion retention [22,38,39]. For instance, in DMD, the composition and volume of secretion are globally normal, but the cough effectiveness is altered in the inspiratory and expiratory cough phases due to muscular weakness. Indeed, the inspiratory phase is impaired due to diaphragmatic weakness which prevents high lung volumes to be reached. The expiratory phase is the most affected because, in addition to rib cage expiratory muscle weakness, the scoliosis adds supplemental mechanical disadvantages [40]. Expiratory muscles are also significantly weakened in some patients with FSHD, but an alteration of the ability to clear secretions from the lung has not been shown [34]. Ineffective cough also decreases the clearance of upper airways favoring benign upper-respiratory infection [41]. However, in association with a swallowing impairment due to the bulbar upper airway muscle dysfunction, the stagnation of secretion at this level leads to their chronic aspiration in the bronchi and the recurrence of pulmonary infection [21,22,42].

On the other side, ventilation is also frequently compromised in MDs due to respiratory pump failure. Respiratory muscle contraction generates a change in airway pressure allowing the movement of air through an open airway, increasing or decreasing lung volume depending on the activated muscles. This pressure generating ability is influenced both by the force generated by the muscle and by the elastic properties of the respiratory system. The elastic recoil of the respiratory system can be appreciated by measuring its compliance, that is its change in volume per unit change in the pressure exerted on it (and is the sum of the compliance of the chest wall and of the lung). When MDs affect respiratory muscles, their pressure generating ability decreases (usually called respiratory muscle weakness), reducing their potential to inflate or deflate the lungs. In addition, MDs also reduce the compliance of the respiratory system increasing the load against which the respiratory pump must operate [22]. Factors which increase the respiratory load involve low lung compliance and altered elastic properties of the chest wall. Causes of the reduced lung compliance are unclear. Some hypotheses have been proposed such as an incomplete maturation of lung tissue, micro- or macro-atelectasis (alveolar collapse) induced by hypoventilation, increase in alveolar surface tension and fibrosis. Concerning the increased stiffness and the decreased compliance of the chest wall, a combination of factors including muscle atrophy and osteoporosis (both consequences of inactivity), extra-articular contractures, progressive degeneration of articular cartilage and last but not least kyphoscoliosis could contribute to this phenomenon [1,21,22]. For instance, severe reductions in the chest wall compliance have been reported in patients with DMD and seem to be secondary to scoliosis. Indeed, statistically these patients lose 4% of their forced vital capacity (the maximal volume of gas that can be in- or ex-haled from the respiratory system) per year with a supplemental 4% decrease for each scoliosis severity degree reached [43]. In LAMA2 Muscular Dystrophy (MDC1A), respiratory involvement is characterized by a progressive restriction of the chest wall involving weakness of the intercostal and accessory muscles. In early stage, thoracic stiffness appears and chest wall compliance decreases [44]. Altogether, lung and chest wall compliance defect increase the muscle force needed to change lung volume. The respiratory work of breathing is therefore increased. As disease evolves, an imbalance appears between the load that has to be overcome and the capacity to overcome it, contributing to respiratory muscle fatigue and, ultimately, to respiratory failure [22]. 

The main consequence of respiratory failure is hypoxemia. The resulting cellular hypoxia can be divided into two forms determined by the dynamics of oxygen deprivation. First, chronic hypoxia (CH), is mainly caused by lung failure and characterized by low oxygen saturations for prolonged periods. Second, chronic intermittent hypoxia (ChIH), usually associated to obstructive sleep apnea syndrome (OSAS) describes transient O_2_ reduction followed by reoxygenation periods [17,45]. Upper airway and respiratory muscle weakness are both involved in sleep-related breathing disorder (SRBD) in MDs. A high body mass index (BMI), tongue muscle weakness (especially in DM [46]) as well as craniofacial abnormalities favoring anatomical defects, can predispose patients to upper airways obstruction and OSAS during sleep. During REM (rapid eye movement) sleep and in NREM (non-rapid eye movement) deep sleep, upper airway resistance increases because the tone of skeletal muscles such as pharyngeal dilator falls. The resulting obstruction impairs airflow through the airways. The direct consequence is a desaturation leading to micro-awakening allowing reoxygenation: multiple repetitions of this sequence cause sleep fragmentation and daytime sleepiness [18,22]. On the other hand, the reduction of muscle tone also affects inspiratory muscles. While this reduction has a limited effect in normal subject, in MDs with muscle weakness and decreased compliance of the respiratory system, it induces a reduction of the tidal volume, contributing to nocturnal hypoventilation. In addition, lying in a supine posture contributes to this hypoventilation by placing the weakened muscles in an adverse mechanical condition (resulting in early symptoms of orthopnea). Four stages have been described concerning the evolution of respiratory failure in these patients: (1) SRBD, (2) hypercapnia (PaCO_2_ above 45 mm Hg) and/or hypoxemia (PaO_2_ lower than 60 mmHg) only during sleep REM phase, (3) hypercapnia and/or hypoxemia also during NREM sleep phase, and ultimately (4) diurnal chronic respiratory insufficiency [18]. According to the MD form, the onset of respiratory insufficiency can be subtle and underdiagnosed, underlining the importance of an early respiratory monitoring and management of SRDB in patients with MDs. Nocturnal oxygen desaturation is indeed associated with a worse prognosis [47]. In FSHD, respiratory involvement has first been considered as a rare manifestation of the disease associated to severity. A study reported a very low prevalence in patients with only 1% requiring home ventilatory support [48]. However, increasing evidence suggests that a nocturnal respiratory support is necessary in a larger proportion of them (around 40%). Notably, a study involving 31 patients reported that half of them presented OSAS [20,49,50]. SRBDs were also reported in DM, even in the absence of typical OSAS symptoms such as excessive daytime sleepiness, highlighting the need of a polysomnographic evaluation in those patients [51].

### 2.2. Muscle Ischemia 

Muscle ischemia is a condition encountered in MDs associated to vascular dysfunction and altered angiogenesis. The inadequate supply of nutrients and oxygen to the muscles is associated to a decreased removal of waste products [52]. HIF-1α is stabilized during ischemia, leading to transcriptional activation of target genes that lead to vascular growth such as those encoding vascular endothelial growth factor (VEGF) and endothelial nitric oxide synthase (eNOS) [53]. 

Duchenne and Becker muscular dystrophy (DMD and BMD) are X linked disorders involving devastating muscle wasting. Both DMD and BMD are caused by mutations in the gene encoding dystrophin, a cytoplasmic cytoskeletal protein localized below the sarcolemma that it stabilizes. Dystrophin is also a scaffolding protein that acts as a mechanical link between structural and signaling proteins, forming a large and highly organized glycoprotein complex called DGC (dystrophin-associated glycoprotein complex) [54]. The muscle-specific isoform of neuronal nitric oxide synthase (nNOSμ) is part of this complex. NO produced by this sarcolemmal nNOSμ normally acts as a local paracrine signal that enhances blood flow in the active muscles by decreasing sympathetic vasoconstriction. This protective mechanism called functional sympatholysis, is impaired in DMD/BMD and results in a muscle ischemia because sympathetic vasoconstriction is not inhibited [55]. Indeed, in DMD, dystrophin deficiency causes nNOSμ mislocalization, leading to the reduction of the paracrine signaling from muscle-derived NO to the microvasculature, which makes the muscle fibers more susceptible to functional ischemia during exercise. Moreover, a reduction in capillary density has been reported both in patients with DMD and in the mdx mouse model (spontaneous non-sense mutation in exon 23 of the dystrophin gene [56]), associated to an enlargement of the remaining capillaries [57,58]. Another study showed that angiogenesis was impaired in mdx mice [59]. Taken together these data suggest that DMD is associated to a defect in vascular function and angiogenesis, and that the muscles are undergoing an ischemic condition. Moreover, mdx mouse satellite cells (SC) present lower angiogenic capacity, as shown by their decreased *Vegf* expression. This decrease appears linked to a lower *Hif-1α* expression in SCs [60]. Therefore, angiogenesis is currently considered as a novel therapeutic target for DMD [61,62] and administration of angiogenic factors, including VEGF and FGF, has been tested as potential treatment to enhance vascularization in ischemic diseases [63]. According to the authors, targeting HIF-1α, which is known to have pro-angiogenic activity, may represent a superior therapeutic approach due to the multiple pro-angiogenic pathways it controls. Indeed, blood vessels formed in pathological conditions (such as in tumor or hypoxic wounds) typically are tortuous, and leaky. This abnormal phenotype is often observed experimentally when inducing angiogenesis with a single agent, such as VEGF and as in *Vegf* overexpressing mice [64]. On the other hand, mice expressing constitutively active forms of Hif-1α and Hif-2α are also hypervascular, but present normal blood vessel phenotype [65].

In FSHD, a histopathological study on patient muscle biopsies suggested a decrease of capillary density [66]. Interestingly, patients affected with infantile FSHD can present exudative retinopathy due to retinal telangiectasias [67,68]. Moreover, as mentioned above, the pathways of HIF-1α and hypoxic signaling response [69,70,71,72,73] as well as angiogenesis were found upregulated in FSHD muscle gene expression profiles [74]. Such deregulations may be involved in mechanisms underlying retinal blood vessel disorder in FSHD. However, the link between DUX4, the causal gene of FSHD, the microvascular abnormalities and the consequences of this capillary density defect (e.g., ischemia) remain unclear. In addition to DMD and FSHD, muscle capillary alterations were also reported in other MDs such as dystroglycanopathies. In Fukuyama type congenital MD, vascular modifications were described such as vessel basement membrane replication, blister-like swelling of endothelial cells, and platelet adhesion and aggregation in small blood vessels [75]. Capillary impairments in the eyes were also reported in another congenital MD called muscle–eye–brain [76,77].

Vascular alterations in MDs could thus be responsible for the activation of hypoxic response pathways in skeletal muscle due to lack of oxygen supply associated to ischemia. This could then lead to HIF-1α stabilization in skeletal muscle whose consequences are discussed below (Section 3).

### 2.3. MD Primary Genetic Defect 

As described above, HIF factors are stabilized under hypoxia in healthy tissues where they promote physiological adaptive mechanisms [78]. However, hypoxia and HIF-1α activation were also shown involved in pathological mechanism and have been particularly studied in the cancer field. Detailed mechanisms were previously reviewed in [79]. Interestingly, an aberrant induction of HIF-1α and HIF-2α expression or stabilization were also reported independently from hypoxia. This condition, named “pseudohypoxia”, was also mostly described in cancer, notably associated to mutations in the gene encoding pVHL [80]. Moreover, recent data highlighted such pseudo-hypoxic patterns in some MDs. 

Indeed, recent studies identified the HIF-1α pathway as critically disturbed in FSHD. FSHD is characterized by a progressive and often asymmetric skeletal muscle weakness. The underlying molecular mechanism is complex and involves both genetic and epigenetic components leading to the activation in skeletal muscle of DUX4 (Double Homeobox 4), a gene normally only expressed in germline and early embryogenesis. DUX4 encodes a potent transcription activator that has a causal role in FSHD pathophysiology [81,82,83]. Several DUX4 direct targets were identified in murine or human myoblasts but the complete DUX4 network is not completely known and mechanisms by which its stochastic expression in very few myonuclei leads to muscle weakness still have to be clarified [84,85]. Meta-analyses integrating gene expression data with known protein interactions identified the hypoxic response pathway as one of the main rewired networks in FSHD muscle biopsies [69,70]. These data confirmed previous transcriptomic studies that had described HIF-1α-signaling as one of the over-represented pathways among FSHD dysregulated genes [71]. Additional confirmation recently came from a genome-wide CRISPR-Cas9 screen performed to identify genes whose loss-of-function would allow survival of muscle cells expressing DUX4. Several genes of the hypoxia response pathway were found as drivers of DUX4-induced cell death [72]. Recent data mining studies have found that besides DUX4 target gene activation FSHD muscle biopsies presented a specific inhibition of the target gene signature of the muscle-specific transcription factor PAX7. Of interest, HIF-1α gene expression is normally inhibited by PAX7 and PAX7 inhibition could thus contribute to HIF-1α increase in FSHD muscles [70,86] (reviewed in [87]). In addition to the primary genetic defect, and a putative direct impact of DUX4 on HIF-1α signaling, additional indirect activation could occur since a subgroup of patients with FSHD also present respiratory insufficiency. 

HIF-1α was also found deregulated in other muscular disorders such as DMD but this activation is probably an indirect consequence of other pathophysiological processes such as inflammation, oxidative stress, angiogenesis and muscle regeneration. These points are discussed below (Section 3). 

## 3. Consequences of Hypoxia and HIF-1α Pathway Activation on Skeletal Muscle

Skeletal muscles are continuously requiring oxygen supply to ensure their functions in motion, postural stabilization and breathing. They consume an important proportion of the whole-body oxygen uptake. If the local oxygen needs exceed the available supply, hypoxic stress occurs and compensatory mechanisms are activated which are collectively named hypoxia stress response pathways. The physiological activation of these pathways leads to the induction of multiple effector genes that modulate various cellular processes such as glucose metabolism and oxidative stress [17,78,88,89]. In skeletal muscle, hypoxia has been shown to modulate not only muscle fiber type profile but also myogenesis and regeneration [90,91,92]. 

### 3.1. Impact on Myogenesis and Regeneration 

Along with their high plasticity, skeletal muscles possess a very efficient regenerative capacity. Only weeks after a complete destruction of fiber integrity, skeletal muscle structure and function can be completely restored [93]. The skeletal muscle regeneration process is carried out by several cellular and molecular events that lead to the restoration of muscle mass, muscle vascularization, and innervation, as well as the recovery of its contractile function [94]. It is important to underline that even if skeletal muscle regeneration shares some similarities with embryonic myogenesis [90,91], they differ in some aspects of their regulatory processes. Indeed, genetic requirements, e.g., their dependency on myogenic factors such as *Myogenin* and *PAX7*, are different for embryonic, fetal, post-natal and adult regenerative myogenesis [95,96,97]. The main mediators of muscle regeneration are myogenic progenitors called satellite cells (SCs) localized between the sarcolemma and the basal membrane of muscle fibers. Under resting conditions, SCs are quiescent and express markers such as *PAX7* and *MYF5*. After muscle injury, SCs are activated by various signals coming from the damaged area. Activated SCs (*MYF5*+) then migrate toward the injury site and begin to proliferate by symmetric division. A subset of SCs (*MYF5*−) can undergo self-renewal to replenish the SC pool either through symmetric or asymmetric cell divisions. At this stage, these cells are called myoblasts and express the myogenic markers *PAX7*, *MYF5*, and *MYOD*. After the proliferation phase, myoblasts differentiate into mature myocytes characterized by a decreased expression of *PAX7* and *MYF5*, and by an increase in *MYOG (**Myogenin-encoding gene*) and *MRF4* expression. Finally, myocytes fuse either to form multinucleated myotubes or to repair the damaged myofibers (for review see [91]) (Figure 3). In addition to the central role of SCs, muscle regeneration is a highly regulated process involving the coordinated action of additional cell types, including fibro-adipogenic progenitors, endothelial cells and macrophages [98]. 

Stem cells in both embryonic and adult organisms frequently reside in hypoxic micro-environments called hypoxic niches. A reduced oxygen level seems critical for the regulation of stem cell activity including their self-renewal, proliferation, and differentiation. Myogenic progenitor cells are also present in a hypoxic environment during embryonic development. Indeed, human development is characterized by a “physiological hypoxia” inducing a HIF-dependent transcriptional coordination of numerous genes [102]. Moreover, HIF-1α is involved in mechanisms governing the quiescence of tissue-resident stem cells [103]. Since SCs are considered skeletal muscle progenitors, it is reasonable to hypothesize that the oxygen level and HIF-1α pathway can affect their activity during embryonic myogenesis and adult regeneration processes [99].

In normoxic conditions, Hif-1α can be detected in skeletal muscle but its protein level is dependent on the muscle fiber type [104]. A recent study in vivo showed that Phd2-deficency and the subsequent Hif-1α accumulation in mice enhanced and accelerated skeletal muscle regeneration after a myotrauma [100]. In this model, the authors also described an accelerated macrophage recruitment to the injured area [100]. Myotrauma was also reported to induce a hypoxic microenvironment leading to Hif-1α accumulation in myofibers and myeloid cells. In muscles of myeloid *Hif-1α* KO mice, myoblast proliferation, regenerating fiber growth and macrophage invasion were delayed after trauma [105]. In addition, Hif-1α silencing in C2C12 muscle cells or its chemical inhibition by echinomycin, significantly altered the differentiation process as shown by the decrease of *Myog* and *Myosin heavy chain* (*MHC*) expression [101]. Taken together, these studies suggest that HIF-1α is necessary for myogenic differentiation under physiological conditions (Figure 3). 

The influence of oxygen levels on myoblast differentiation into myotubes in vitro has been widely studied but many discrepancies exist among publications in the field. First, the effect of hypoxia on myoblast differentiation in vitro appears to be strongly dependent on the depth of hypoxia. Indeed, several publications have demonstrated that a low oxygen level (O_2_ ≤ 1%) had harmful effects on myogenic differentiation [106,107,108,109,110,111]. For instance, the use of a chemical hypoxia-mimicking agent, namely cobalt chloride, inhibited C2C12 myoblast differentiation in a dose-dependent manner through downregulation of the myogenic factor Myogenin [112] while a moderate hypoxia could promote differentiation of C2C12 muscle cells [113]. However, depending on the dose of chemical compound used to stabilize HIF-1α and on the studied cell line, HIF-1α-independent side effects might be observed such as apoptosis, disruption of the mitochondrial transmembrane potential, upregulation of the voltage-dependent anion channels [114]. Hypoxia (1% O_2_) promoted the differentiation of bovine primary myoblasts whereas the differentiation of C2C12 myoblasts was compromised, underlining the influence of the myogenic lineage type [115]. To summarize, the effects of hypoxia on myogenic differentiation are not completely understood, and discrepancies between studies are likely related to the different experimental parameters used such as the duration, the depth and the type of hypoxia (chemical, normo/hypobaric) as well as the myogenic lineage used (e.g., immortal cell line vs. primary cells, species (e.g., mouse vs. human), culture media) (Table 3). Finally, it is important to underline that cell cultures are usually performed in a humidified 95% air atmosphere, supplemented by 5% CO_2_, providing about 20% O_2_. In such hyperoxic environment, cells were shown to reset their normoxic set-point by downregulating PHD [116]. Those conditions, thus commonly considered as “normoxia” are not representative of O_2_ partial pressure in tissues in vivo that results from the balance between oxygen supply and consumption. In skeletal muscle, the physiological range of O_2_ level (termed “physioxia”) is significantly lower than 20% (4% O_2_) [117,118,119,120]. The interpretation and translation to whole organisms of data obtained in vitro have to be made with caution by taking into account that cellular and molecular reactions to hypoxia may differ from those occurring in vivo.

Since studies have shown that the oxygen level regulated myoblast differentiation into multinucleated myotubes in vitro [106,107,108,109,110,111,112,113,115], the next question was whether hypoxia could affect skeletal muscle regeneration in vivo. In a model of *soleus* muscle injury induced by notexin in rats, prolonged hypobaric hypoxia (28 days at 10% FiO_2_ (fraction of inspired oxygen)) was found to repress the early regeneration process. This repression could be linked to an attenuation of the increase of *MyoD* and *Myog* expression by hypoxia during the first week of regeneration [121]. In another study hypoxia was mimicked during skeletal muscle regeneration in rats by using dimethyloxalylglycine (DMOG) to stabilize Hif-1α in injured fibers. DMOG induced a defect in the activation of the myogenic factor genes *Myf-5* and *Myog* [122]. Altogether, those data suggested that contrary to Hif-1 stabilization, notably through Phd2 deficiency, severe and prolonged hypoxia as well as the chemical hypoxia-mimetic DMOG had a negative impact on myogenic differentiation during muscle regeneration in vivo [122,123,124]. 

Dystrophic muscles are characterized by an altered regeneration capacity along with chronic inflammation and fibrosis. Data pointing towards SC implication in MDs are accumulating, but SC contribution to muscle pathophysiology is not precisely understood [125]. Interestingly, the number of SCs was found similar in FSHD and healthy muscles [66]. Active regeneration was demonstrated in FSHD muscle biopsies by two criteria: (i) increased transcriptional expression of regeneration markers consisting of 200 human genes associated with myogenesis; (ii) presence of regenerating fibers as shown by immunolabeling for developmental myosin heavy chain [126]. However, the progressive muscle wasting observed in patients implies that this regeneration is not sufficient to prevent dystrophic changes. In another study, the proteomic profile of interstitial fluids in patient muscles showed a downregulation of structural muscle proteins and of the plasminogen pathway. Along with the inhibition of myogenic factors, this study suggested that muscle regeneration was impaired in FSHD, along with an increased fibrosis [127]. Hypoxia and HIF-1α were found involved in the establishment of muscle fibrosis through crosstalk with profibrotic factors, such as Transforming Growth Factor β (TGF-β) [88]. In DMD, muscles undergo repeated cycles of degeneration and impaired regeneration resulting in muscle wasting, fibrosis and fat accumulation [128]. A study focused on DMD and using mRNA profiling with large-scale data integration found that TGF-β–centered networks strongly associated with fibrosis and regeneration alteration. It also highlighted HIF-1α as a notable component of this network [129,130]. Besides their role in muscle regeneration, SCs contribute to the revascularization of damaged muscles by secreting angiogenic factors such as VEGF. Moreover, when microvascular fragments (composed of endothelial, pericyte, and smooth muscle cells) were co-cultured with SCs, they presented stronger angiogenesis capacity than when cultured alone [131]. Another study suggested that Hepatocyte Growth Factor (HGF) played a role in SC-mediated angiogenesis. In the mdx mouse model, *Hif-1α* and *Vegf* expression were found decreased in proliferating SCs from dystrophic muscles compared to wild-type mice. This indicates that mdx dystrophic muscles present a decreased SC angiogenic capacity, partly through mechanisms involving a decreased *Vegf* expression [60] and contributing to the regeneration defect in DMD. Finally, in some MDs, the pathogenic mutation resulting in myofiber wasting also directly impairs SC function and consequently alters the regeneration process. For instance, DMD muscles are characterized by degeneration/regeneration cycles leading to a muscle micro-environment presenting endomysial fibrosis, chronic inflammation and fatty infiltration. Altogether, this creates a hostile niche which could impact SC-mediated repair process [132]. In aggregate, these data demonstrate a key role for HIF-1*α* activation in myogenesis and healthy muscle regeneration. We could therefore hypothesize that either low or excess HIF-1*α* activation could contribute to MD muscle pathology by limiting regeneration or favoring fibrosis.

### 3.2. Ultrastructural Modification

Skeletal muscles are composed of different fiber types classified according to their metabolic and contractile properties. Slow-twitch oxidative (SO) fibers referenced as type I are rich in mitochondria, myoglobin (involved in oxygen storage) and are characterized by a high capillary density. They have a strong resistance to fatigue and rely predominantly on oxidative phosphorylation to produce their energy. Fast-twitch glycolytic (FG) fibers referenced as type II show low mitochondrial content and are more prone to fatigue. Finally, fast-twitch oxidative glycolytic (FOG) fibers are characterized by intermediate properties [133]. Over the past 25 years, modifications in the composition of myofiber types were described in conditions of pathological hypoxia, such as obstructive sleep apnea syndrome (OSAS) and respiratory failure observed in subgroups of patients with MDs [22]. A slow-to-fast fiber-type transition occurred in the lower limb muscles during chronic obstructive pulmonary disease (COPD) and in upper airway muscles during OSAS. Several factors were suggested to explain this transition.

In physioxia, Hif-1α is significantly expressed and stabilized in skeletal muscle but *Hif-1α* gene expression and protein level depend on the muscle fiber type. Indeed, higher *Hif-1α* mRNA and protein levels were detected in predominantly glycolytic muscles, namely the *gastrocnemius* and *quadriceps*, as compared to oxidative muscles such as the *soleus* [104]. The authors linked these Hif-1α protein variations with the myoglobin content of myofibers that is higher in oxidative muscles than in glycolytic ones. It is important to remind that skeletal muscle myoglobin acts as an intracellular oxygen buffer and constitutes an oxygen reservoir even at low PO_2_. Hif-1α was found critical in a slow-to-fast myofiber switch, but some discrepancies remain among studies. In transgenic mice with a skeletal muscle-specific *Hif-1α* gene inactivation, the proportion of type-IIa fibers was slightly reduced in *soleus* muscles as compared to the control mice and thus suggests a slightly slower fiber-type profile. Interestingly, fiber-type profile from *gastrocnemius* muscles did not vary between the two groups. This study also showed that endurance performance was better in *Hif-1α* KO mice than in wild-type mice. However, repeated exercise bouts induced more severe muscle damage in the KO mice, which consequently impaired their running performance after four consecutive days of exercise training [134]. Furthermore, the fast-to-slow fiber-type shift and the enhancement of oxidative capacity induced by long endurance training were impeded by *Hif1* KO in skeletal muscle [135]. In contrast, Hif-1α stabilized by *Phd2* conditional muscle KO caused a shift toward a slow fiber type via a calcineurin/Nfatc1 signaling pathway in the *soleus* and *gastrocnemius* muscles [136]. Accordingly, *Phd2* KO mice had better endurance performance after training compared with control mice [137]. Interestingly *Phd2* gene inactivation could stabilize Hif-2α as shown by its accumulation in *Phd2* null mouse muscles. However, this effect cannot be only attributed to the myofiber type switch. Indeed, *Phd2* KO mice also presented an upregulation of plasmatic erythropoietin (Epo) level associated to an increased hematocrit and that likely participated to enhance aerobic capacity and endurance. Since *Hif-2α* KO leads to a slow-to fast fiber-type switch in murine *soleus* muscles [138], Hif-1α and Hif-*2α* could therefore have differential roles in the determination of the contractile phenotype and in adaptation to exercise. Although both factors share an identical core DNA binding motif, Hif-2α was considered less crucial than Hif-1α for acute induction of HIF target genes, but some evidence suggests that Hif-2α exerts its influence in long-term exercise adaptation. Hif-1α is known to be stabilized in response to an acute bout of endurance exercise but seems to be repressed by long-term endurance exercise through induction of its negative regulators [14]. This inhibition might be necessary for the switch to oxidative metabolism which is critical for endurance exercise adaptation of the skeletal muscle. In addition, Pgc1α activation by endurance exercise training specifically induced *Hif*-2α expression. This activation was dependent on estrogen-related receptor α [138]. The *Pgc1α/Hif-2α* link was confirmed by the observation that *Hif-2α* expression was decreased in the *gastrocnemius* of muscle-specific *Pgc1α* KO mice. Concomitantly, Hif-2α induces the transcription of genes associated with slow-twitch oxidative muscle fiber phenotype and seems critical for the Pgc1α-induced oxidative switch in vitro [138]. Altogether, this shows that Hif-2α is a downstream part of the Pgc1α pathway, known to induce mitochondrial biogenesis, and acts as a key regulator of a muscle fiber-type program and the adaptive response to endurance exercise. 

Fiber types are not affected in the same way in all MDs [139]. In DMD, destruction of type II muscle fibers is an early event while type I muscle fibers are lost at later stages [140]. A similar trend was observed in FSHD muscles where an early loss of type II fibers is observed together with an overall increased proportion of type I fibers [141]. In contrast, patients with DM initially present atrophy of type I fibers [142]. It remains unclear why specific fiber types are affected in certain MDs. Thus, understanding this difference of disease sensitivity may provide important insights into the pathophysiology and the development of treatments. For instance, since type II fibers appear initially affected in DMD, it was proposed to selectively promote slow muscle fiber function as a potential therapy to delay DMD progression [143]. Transgenic overexpression of *Pgc1α* resulting in a slower fiber type phenotype [144] was shown to ameliorate muscle structural and functional defects in the mdx mouse [145]. These studies provide potential candidates that could be tested as therapeutic for the rescue of muscle dysfunction in DMD. 

Altogether, available data highlight that in MDs, both the primary genetic defect and hypoxia could differently modulate myofiber type distribution, a key element of muscle function and physical properties such as fatigue resistance and exercise tolerance. A better understanding of factors and mechanisms involved in those changes remain essential to provide insights for MD patient care. In this context, the HIF-1α pathway remains a central target to investigate given its role in fiber type regulation. 

### 3.3. Metabolic Alterations

HIF-1α is known as an important mediator of metabolic changes occurring under hypoxia. Interestingly, hypoxic response pathways are physiologically activated under particular conditions e.g., in stem cell quiescence and cell torpor (hibernating animals) [103]. In these “dormant” conditions, quiescent cells have to switch their metabolism from oxidative phosphorylation (OXPHOS) to fatty acid oxidation, anaerobic glycolysis, glutaminolysis and pentose phosphate pathway in the aim to secure a minimal energy supply and avoid metabolic dysregulation and oxidative stress [103,146,147]. HIF-1α was reported to play a key role in those metabolic adaptations. Moreover, HIF-1α is involved in metabolic changes occurring under pathological conditions, with most studied examples in the cancer field. Indeed, solid tumors are often exposed to hypoxic micro-environments. In order to support their growth and proliferation, cancer cells alter their metabolism by downregulating OXPHOS and increasing aerobic glycolysis. This metabolic shift is called “Warburg effect” and enables a rapid ATP generation to the detriment of large amounts of glucose consumption (reviewed in [148]). It also allows for NAD^+^ production through pyruvate to lactate conversion.

Concerning skeletal muscle metabolism, we first have to mention that muscle activity is critically dependent on oxygen supply to maintain both energetic and redox status. Indeed, ATP hydrolysis provides an immediate energy source, but intramuscular stores of ATP are very limited and rapidly consumed. Therefore, metabolic pathways driving ATP generation are necessary to meet skeletal muscle energy requirements. Mitochondrial OXPHOS provides most ATP molecules in normoxic skeletal muscle. OXPHOS involves the reduction of oxygen to water with electron transferred from reducing equivalents (NADH and FADH_2_) that are generated in catabolic pathways such as glycolysis and β- oxidation of fatty acids [149].

Muscle metabolic adaptations occurring upon exercise are highly dependent on training modalities, duration, frequency and intensity as reviewed in [150]. For short term energy supply, ATP is derived from phosphocreatine stock via the Lohmann reaction which takes place in the cytosol. In case of prolonged exercise, ATP will be provided by anaerobic glycolysis in the cytosol and by mitochondrial OXPHOS. Anaerobic glycolysis produces pyruvate that is then converted into lactate by lactate dehydrogenase in a reaction that provides NAD^+^ electron acceptor needed for glycolysis. 

Similarly, in experimental studies in vitro, exposure of skeletal muscle cells to hypoxic conditions leads to HIF-1α activation and the subsequent up-regulation of 11 genes encoding glycolytic enzymes (aldolase A, aldolase C, enolase 1, glyceraldehyde-3-phosphate dehydrogenase, hexokinase 1, hexokinase 2, lactate dehydrogenase A, phosphofructokinase L, phosphoglycerate kinase 1, pyruvate kinase M, and triosephosphate isomerase), promoting a glycolytic metabolism [151] (Figure 4). In contrast, studies on healthy individuals exposed to environmental hypoxic conditions did not measure any increased activity of most glycolytic enzymes in skeletal muscle (as reviewed in [152]). This difference could come from experimental conditions e.g., the hypoxia exposure pattern. However, besides its activity on glycolysis, HIF-1α reduces OXPHOS through the induction of pyruvate dehydrogenase kinase 1 (PDK1) which decreases pyruvate entry into the mitochondrial Krebs cycle. Indeed, PDK1 inhibits by phosphorylation the E1α subunit of the pyruvate dehydrogenase enzymatic complex that converts pyruvate to acetyl-coenzyme A and CO_2_ in the mitochondria. In skeletal muscle cells, Pdk1 upregulation was observed in rats exposed to 10% FiO_2_ for 2 weeks [153]. Accordingly, the same upregulation was observed in human myoblasts and in mouse C2C12 cells treated with a Phd inhibitor [154]. C2C12 muscle cells treated with a Phd inhibitor presented increased production of lactate [154]. Consequently, based on cell culture studies, lactate production was expected to increase in muscle with hypoxia exposure. However, some studies did not find variation in lactate concentration and lactate-to-pyruvate ratio in muscles of healthy subjects submitted to environmental hypoxia at high altitude [155,156]. This observation highlights complex skeletal muscle regulations in vivo. The impact of hypoxia on carbohydrate oxidation is better documented than its consequences on lipid metabolism. In the liver, HIF-1α and HIF-2α are involved in hypoxia-induced lipid accumulation via a reduced fatty acid β-oxidation [157]. However, in skeletal muscle, “AltitudeOmics” studies on muscle biopsies of healthy volunteers revealed that adaptation to hypoxia was probably more complex than a simple shift from aerobic to anaerobic metabolism. According to these authors, a more efficient fatty acid β-oxidation may participate in an early phase of high-altitude hypoxia adaptation, by providing reduction equivalents to the OXPHOS electron transport chain in the absence of those derived from glycolysis [158]. Moreover, as mentioned earlier, skeletal muscle myoglobin can act as a buffer of intracellular oxygen concentration and constitutes an extra reserve of oxygen even at insufficient PO_2_ as encountered in hypoxic conditions. Therefore, this oxygen stock could contribute to the maintenance of ATP production by fatty acid β-oxidation in mitochondria, at least in early hypoxia adaptation.

Alongside with this metabolic remodeling, hypoxia exposure leads to a decrease of mitochondria content via HIF-1α-induced mitochondrial autophagy [159]. Autophagy is a catabolic process that eliminates or recycles defective proteins and cytoplasmic organelles to maintain homeostasis in the cell or as a mean of providing macromolecules for energy production under starvation conditions [160]. Autophagy involves the formation of autophagosomes which is initiated by the dissociation of the Beclin1/Bcl-2 complex. HIF-1α mediates an increase in Bnip3 expression during hypoxia exposure in skeletal muscle [161,162,163,164]. Bnip3 competes with Beclin1 for Bcl-2 binding which facilitates Beclin1 release and autophagy (Figure 4) leading to decreased metabolic contribution of mitochondria and increased involvement of the glycolytic pathway to produce energy. Accordingly, mice with selective inactivation of *Hif-1α* gene in skeletal muscle showed an increased oxidative capacity and mitochondrial content, a reduction of lactate concentration in serum and an enhanced performance during training [134]. 

Many MDs are associated with muscle metabolic defects. Studies have shown disturbances in glucose metabolism, including reduced glycolytic substrates, glycolytic enzyme activity such as lactate dehydrogenase, aldolase, and pyruvate kinase, and defects in insulin receptor signal transduction in DMD muscle biopsies, supporting the hypothesis of a reduced glycolytic activity [165]. Interestingly, *EDL (Extensor Digitorum Longus)* muscle from utrophin-dystrophin deficient dystrophic mice presented an increase in Hexokinase 1 (Hk-1) and Pyruvate kinase M2 protein levels [166]. This can be explained by the presence of more regenerating fibers with proliferating cells which mainly rely on glycolysis. Another study on golden retriever muscular dystrophy (GRMD) highlighted reduced expression of glycolytic enzymes such as 1, 6-phosphofructokinase which is regulated by Hif pathway [167]. Moreover, an mRNA profiling performed on the same model showed *Glut4* downregulation but increased *Hk-1* expression [168].

It is believed that the mitochondrial functional changes observed in DMD are mainly linked to dysregulation of Ca^2+^ homeostasis. Indeed, an early decrease in the efficiency of Ca^2+^ transport and accumulation in mitochondria was reported in mdx mice and associated with a lower rate of mitochondrial OXPHOS [121]. Moreover, evidence of insulin-resistance and other metabolic alterations such as obesity and hyperinsulinemia have been reported in DMD. Abnormal cytoplasmic aggregates of GLUT4 transporter were observed in DMD myofibers suggesting an alteration of glucose uptake in muscles [169]. In DM1, alterations in glucose metabolism, and insulin resistance were reported as an early disease manifestation. Insulin resistance is the main cause of glucose intolerance in DM1 and leads to hyperinsulinemia and later, to diabetes mellitus [170]. A recent study by NMR spectroscopy showed metabolic alterations such as glutamate/glutamine ratio or alanine decreased levels in muscle bioenergetic metabolism of patients with DMD, BMD, FSHD and limb girdle muscular dystrophy ((LGMD)-2B) both in early or acute phases of the disease [171]. Interestingly, evidence for mitochondrial dysfunction was found in FSHD muscle where impaired energy metabolism was associated with alterations in mitochondrial ultrastructure and subsarcolemmal and intramyofibrillar distribution [172]. Furthermore, a dynamic transcriptomic analysis identified that suppression of PGC1α, the co-factor and activator of ERRα, a critical component of the mitochondrial biogenesis pathway, was associated to the myogenesis defect in FSHD [173]. Finally, further understanding could come from the study of mitochondrial myopathies, caused by mutations in mitochondrial DNA (mtDNA) affecting genes involved in OXPHOS such as the electron transport chain. These patients present muscular symptoms including proximal limb weakness, muscle fatigue, exercise intolerance and pain [174].

### 3.4. Oxidative Stress 

Reactive oxygen species (ROS) are reactive molecules and free radicals derived from molecular oxygen such as superoxide anion (O_2_^−^), hydrogen peroxide (H_2_O_2_), hydroxyl radical (OH^.^), hydroxyl ion (OH^−^). Reactive nitrogen species (RNS) are various nitric oxide-derived compounds including molecules such as nitric oxide (NO) or peroxynitrite (ONOO^−^). ROS constitute a double-edged sword: depending on the magnitude, duration and cellular production site and target cells, they can either trigger beneficial or detrimental pathways [175,176]. Indeed, ROS are key signaling molecules in a number of physiological processes including the maintenance of muscle function and adaptation to exercise [177]. However, excessive and sustained ROS production and the imbalance between pro-oxidant and anti-oxidant pathways can cause oxidative damage to nucleic acids, proteins, and lipids that could initiate cell death [178]. In skeletal muscle, the ROS sources are still controversial but several reports indicate that mitochondrial electron transport is strongly involved [179,180,181] (Figure 5 upper part).

It is known that hypoxia favors an increased ROS production both in acute [182,183,184] and long-term hypoxemia [185]. Deleterious ROS effects are highly dependent on the hypoxia intensity and duration [186,187]. Two studies [188,189] have shown that acute hypoxia could lead to diaphragm muscle force reduction, as well as a decrease of both contraction and relaxation times. The use of antioxidant treatments during hypoxia rescued this loss in muscle function. This effect has been shown to be independent from the preservation of high-energy phosphates suggesting that hypoxia-induced ROS inhibits contractile function in a way that is not linked to a loss of energy status of the tissue but to the redox status of the muscle [188]. Moreover, the combination of hypoxia and exercise seems to be involved in the improvement of antioxidant capacity and might influence redox balance in a beneficial way [190]. 

Unlike continuous hypoxia, chronic intermittent hypoxia (ChIH) present in obstructive sleep apnea syndrome (OSAS) has pathophysiological consequences associated not only with hypoxia and hypoxia response but also with oxidative stress as a consequence of fast tissular reoxygenation. Indeed, in OSAS, each episode of intermittent hypoxia is followed by patient micro awakening and breathing, and this re-oxygenation leading to HIF-1α rapid degradation (within 5 min) generates ROS and oxidative stress that contribute to skeletal muscle dysfunction [191].

A previous metabonomic study performed in our laboratory has shown oxidative stress marker imbalance during ChIH exposure in mice [192]. For instance, allantoin, a urine marker of oxidative stress was significantly increased. Urinary taurine and methionine levels were decreased which indicate a higher organism consumption of these antioxidants. Several studies have demonstrated upper airway muscle dysfunction induced by ChIH [193,194,195]. Indeed, cycles of hypoxia-reoxygenation were shown to increase *geniohyoid* and *sternohyoid* muscle fatigue in rat models [196]. Accordingly, pro-oxidants worsen ChIH-induced respiratory muscle dysfunction [193], while the use of antioxidants improves it, underscoring the role of ROS in this phenomenon [193,195,197]. Surprisingly, only few studies focused on the impact of ChIH on locomotor muscles. McGuire et al. reported that chronic intermittent asphyxia led to an increased fatigue in rat *EDL* and *soleus* muscles [198]. Concomitantly, another group exposed rats to chronic intermittent hypoxia-hypercapnia and found a significant downregulation in type I fibers in *soleus* and *gastrocnemius* muscle [199]. Rats exposed to ChIH also presented greater levels of mitochondrial superoxide anion that was significantly reduced by treatment with N-acetyl cysteine (NAC) [200]. 

Although the genetic background of many MDs has been identified, the exact mechanism underlying skeletal muscle dysfunction often remains unclear. Oxidative stress has been deeply investigated in DMD [201] and is also obviously involved in other myopathies such as FSHD [172,202,203,204], SEPN1-related myopathies [205] or laminopathies [206,207]. Indeed, alterations in antioxidant responses including an increased level of oxidized glutathione and higher protein oxidation have been shown in mdx mice and in DMD muscle biopsies [208,209,210]. Moreover, the evaluation of antioxidant drugs in pre-clinical studies performed on mdx mice [211,212,213,214] and in clinical studies on patients with DMD [215,216,217] are supporting the hypothesis of a role of oxidative stress in DMD. Interestingly, a double-blind randomized placebo-controlled phase 3 trial using the short-chain benzoquinone idebenone showed a significant reduction in the loss of respiratory function in DMD [215]. Concerning FSHD, the involvement of oxidative stress in the pathology is supported by both clinical and experimental studies. FSHD myoblasts had an increased susceptibility to oxidative stress in primary culture as shown by their significant decreased viability when exposed to the oxidative stressor paraquat as compared to control [218]. Moreover, FSHD muscle biopsies presented increased lipid peroxidation, protein carbonylation and oxidative damage (e.g., lipofuscin accumulation) as compared to control muscles [172,219]. Concerning the origin of oxidative stress in FSHD, Turki et al. showed that functional muscle alterations were associated with mitochondrial dysfunction [172]. Finally, a randomized, double-blind, placebo-controlled pilot clinical trial has shown that an oral supplementation with an antioxidant combination to complement specific defects observed in FSHD myoblasts could moderately improve muscle function in FSHD patients [202]. Indeed, maximal voluntary contraction, and endurance limit time of the dominant and nondominant quadriceps were significantly improved in the supplemented group. This study also suggested that the effect of supplementation on physical performances might be different depending on antioxidant status and oxidative stress marker baseline of individual patients. Finally, DUX4 gene expression was found increased by oxidative stress and this phenomenon was mediated through the DNA damage response pathway highlighting a vicious circle occurring between DUX4 and oxidative stress [220]. Moreover, a recent study showed that a subset of genes was deregulated by DUX4 indirectly through oxidative stress [221]. Thus, we can hypothesize that in FSHD, oxidative stress could stabilize the HIF-1α pathway which could by itself further contribute to amplify oxidative stress. However, additional studies are necessary to better understand the link between DUX4 expression, mitochondria dysfunction, oxidative stress and HIF-1α activation in FSHD muscles [220]. Recent publications indicated that the target gene signature of PAX7 (an inhibitor of HIF-1α gene expression) was decreased in FSHD muscle cells thus contributing to increased HIF-1α protein levels [87]. Finally, it must be recalled here that OSAS occurring in a subgroup of patients with FSHD could further increase oxidative stress caused by the MD pathology.

Excessive oxidative stress can interfere with processes leading to muscular contractions at different steps (Figure 5 lower part). Indeed, excitation–contraction coupling depends on motor neuron-induced cell depolarization and the subsequent interaction between the dihydropyridine receptor (DHPR) and the ryanodine receptor (RyR1). This leads to the release of Ca^2+^ from the terminal cisternae of the sarcoplasmic reticulum (SR) [222]. RyR1s have been reported as channels sensitive to the redox state of muscle cells. Such alterations can induce their activation or inactivation: oxidative stress results in increased RyR1 opening leading to a Ca^2+^ leak [223]. Alongside with RyR destabilization, a pro-oxidant environment can lead to the activation of Ca^2+^/calmodulin-dependent protein kinase II (CAMKII) which is known to cause RyR1 phosphorylation resulting in a leakiness of Ca^2+^ release from the SR [222]. Finally, ROS can by themselves alter myofilament structure and function. Indeed, myofilament proteins, including myosin and troponin I and C, can be oxidized and present dysfunctions after a long exposure to high ROS levels [224,225,226,227]. Interestingly, in addition to a link with oxidative stress, hypoxia was found to affect Ca^2+^ homeostasis. Indeed, hypoxia significantly decreased the L-type Ca^2+^ channel-dependent Ca^2+^ influx towards the cytosol and prolonged the duration of Ca^2+^ release from the SR through RyR channels [228]. This was confirmed in other cell types (endothelial cells and cardiomyocytes) in which chronic hypoxia increased the levels of cytosolic Ca^2+^ by enhancing its release from the endoplasmic reticulum [229,230]. Oxidative stress and hypoxia could therefore participate in MD muscle dysregulation through Ca^2+^ homeostasis disturbance. Moreover, increased cytosolic Ca^2+^ could activate calpains, a family of calcium-dependent, cysteine proteases. Indeed, mdx mice presented higher amounts and activation of ubiquitous calpains that could contribute to proteolysis and subsequent muscle wasting in DMD [231,232]. In addition, mdx mice had reduced abundance of Csq1 (calsequestrin) and Clp (calsequestrin-like protein) in the heart [233] and skeletal muscle [234], respectively. CSQ1 is the major calcium binding protein in the SR, plays an important role in calcium storage and acts as a regulator of muscle excitation–contraction coupling and stress response. Due to its lower amounts, Csq1 could thus not have a protective role against excess cytosolic Ca^2+^ in mdx mice.

Altogether, current evidence indicates that oxidative stress imbalance in skeletal muscle could contribute to pathophysiological processes in several MDs. We can hypothesize that in MDs presenting such oxidative stress imbalance (either linked to the genetic defect or as an indirect consequence of the resulting muscle pathology), the presence of an OSAS and consequently of a ChIH could exacerbate redox status disturbances and therefore muscle metabolism and dysfunction in those patients. This could be especially the case in patients with MDs with limited antioxidant defenses as reported in FSHD and DMD. This underlines the importance of an early screening and the monitoring of such respiratory problems.

## 4. Pharmacological HIF-1α Modulators in MDs

To our knowledge, pharmacological regulators of HIF-1α are not presently used in MDs. The drugs available to interfere with HIF-1α expression or activity have been developed in the field of cancer and with the final purpose to induce targeted cell death. This obviously has to be avoided in the treatment of muscle disorders. A dose adaptation might allow to suppress muscle cell toxicity. However, several drugs such as those targeting the mTOR pathway seem inappropriate in MDs since they interfere with protein synthesis and would contribute to muscle atrophy [235]. 

Activation or inhibition of HIF pathway components should both be considered in relationship with the type of MDs. In mdx mouse, SCs presents decreased *Hif-1α* and *Vegf* expression that could participate in the reduced angiogenic capacity and regeneration potential. A therapy based on VEGF, an important element of the HIF-1α pathway, has been proposed since alterations in the angiogenesis process have a significant impact on DMD progression. Therefore, direct delivery of VEGF has been suggested as a potential treatment option even if several limitations were highlighted by the authors (severe side effects in case of over-administration, rapid clearance resulting in a need of frequent delivery) [62]. 

Concerning FSHD, hypoxia response was identified as the main contributor to DUX4-induced cell death [72]. In an immortalized myoblast line with inducible DUX4 expression the cellular hypoxia response could be disturbed with inhibitors of the phosphatidylinositol 3-kinase (PI3K)/Akt/mTOR or Ras/mitogen-activated protein kinase (MAPK) signaling pathways [72]. Interestingly, Losmapimod, the first compound currently in clinical trial for FSHD is a p38MAPK inhibitor, and this kinase can regulate HIF1α signaling [236]. Moreover, oxidative stress was found to induce DUX4 expression [219] while the resulting oxidative stress induced additional toxic genes [221]. Agents that reduce oxidative stress allow survival of DUX4 expressing cells [204,237,238] and antioxidant complementation was tested in a clinical trial for FSHD [202]: these antioxidants might also affect HIF-1α which is stabilized by ROS.

## 5. Conclusions

Hypoxia and HIF-1α signaling alterations clearly influence skeletal muscle structure, metabolism, regeneration and function. Both conditions occur in MDs due either to the genetic defect itself (directly or indirectly) or to a resulting respiratory insufficiency or muscle blood vessel abnormalities. Therefore, we can hypothesize that on one hand, hypoxia and MD-associated muscle disturbances themselves may have synergistic effects on key converging processes namely oxidative stress, metabolism and regeneration, initiating therefore a vicious circle whose deleterious consequences could participate in pathophysiological mechanisms underlying muscle weakness in a significant number of patients with MDs (Figure 6). On the other hand, a prolonged and aberrant HIF-1α induction may also occur in MDs independently of hypoxia. Such “pseudohypoxia” could participate in muscle dysfunction in MDs through the activation of improper gene expression programs favoring e.g., cell quiescence or a metabolic shift towards lowered oxygen consumption and ATP production, such conditions being inappropriate for normal function of adult, mature skeletal muscles. 

In this context, a robust monitoring of respiratory function and an early diagnosis of respiratory impairments in patients with MDs constitute key milestones. An appropriate respiratory management including non-invasive nocturnal ventilation should be considered in the presence of early nocturnal signs of hypoventilation. In the aim to provide further recommendations to improve muscle function in MDs, it remains important to increase our knowledge about the influence of hypoxia and HIF-1α molecular mechanisms on MD progression. Current studies seem to point toward HIF-1α as a potential therapeutic target for muscle disorders. HIF-1α pharmacologic regulators and gene therapy tools currently developed in the field of cancer research may thus be useful in the development of multitherapy protocols in MDs, complementary to emerging strategies specific to individual MDs. Personalized medicine should be developed on the bases of precise criteria allowing patient clustering. In this context, a better understanding of HIF pathway components, muscle metabolic profile and redox status resulting from oxidative stress and antioxidant response in MDs will help to define new biomarkers to improve the management of co-morbidities, especially in patients with MD-associated respiratory impairments. 

## Figures and Tables

**Figure 1 ijms-22-07220-f001:**
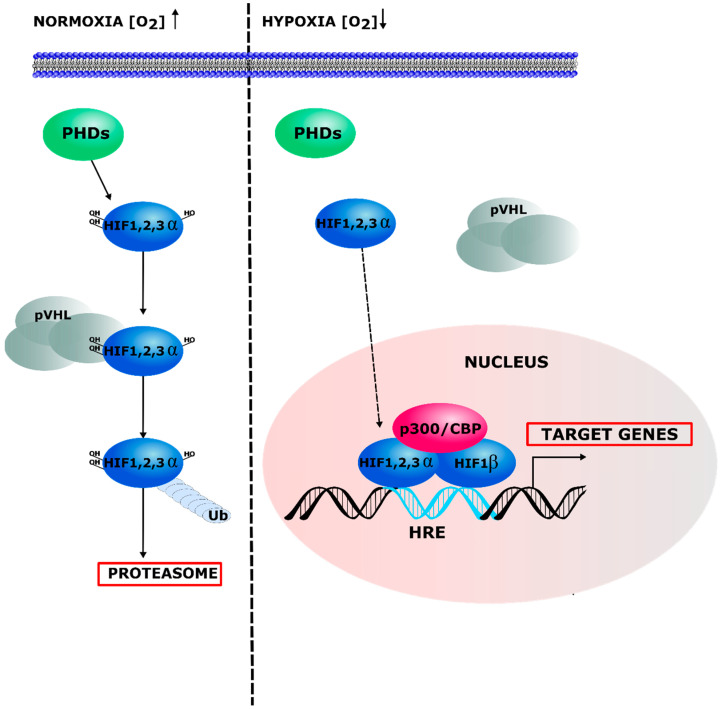
Regulation of HIF pathway. Under normoxia, HIFα protein is hydroxylated by PHDs and subsequently recognized by pVHL and degraded through the ubiquitin–proteasome pathway. In hypoxia, PHD activity is inhibited by the decreased O_2_ levels and thus HIFα hydroxylation. HIFα is therefore stabilized and translocated into the nucleus, dimerizes with HIF1β, and activates the expression of hypoxia responsive genes with additional transcriptional co-factors, such as CBP/P300. Ub = ubiquitin.

**Figure 2 ijms-22-07220-f002:**
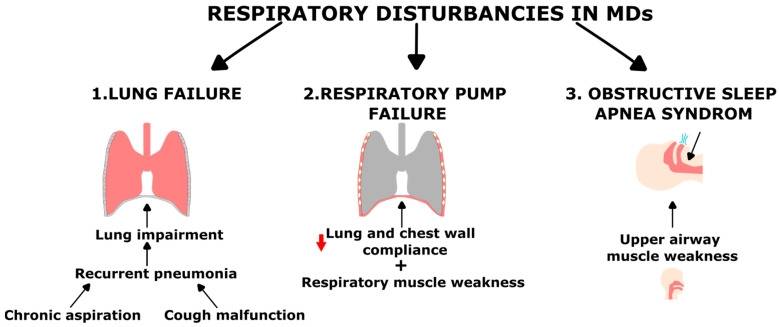
Diagram summarizing the three main causes of respiratory disturbances in MDs. (1) The main cause of lung failure is recurrent pneumonia usually secondary to cough inefficiency or chronic aspiration. (2) Respiratory pump failure is associated to respiratory muscle weakness (decreased pressure generating ability) combined to low respiratory system compliance (increased work of breathing). (3) Obstructive sleep apnea syndrome (OSAS) is also a common respiratory impairment, weakness of upper airway muscle enabling the inspiratory airway closure.

**Figure 3 ijms-22-07220-f003:**
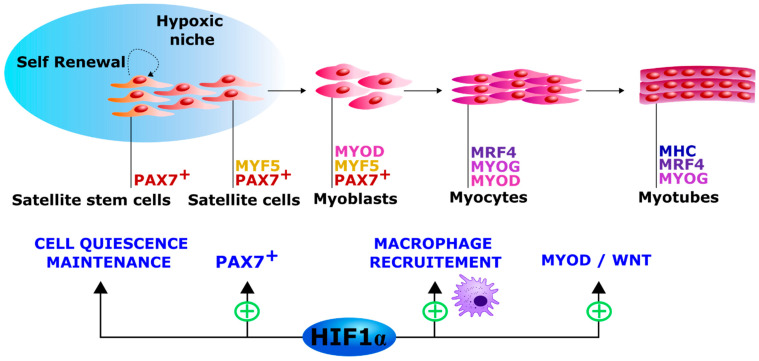
Myogenic differentiation. SCs are myogenic progenitors and are localized in hypoxic niches near the muscle basal membrane. Following muscle injury, quiescent SCs (*PAX7+* and *MYF5+/−*) are activated and differentiate into myoblasts (*PAX7+*, *MYF5+*, and *MYOD+*). After their proliferation cycle, myoblasts differentiate into myocytes (*PAX7−*, *MYOD+*, *MYOG+*, and *MRF4+*). Finally, myocytes fuse together to form multinucleated myotubes (*MYOG+*, *MRF4+*, and *MHC+*). The satellite stem cell subpopulation (*PAX7+* and *MYF5−*) can proceed to self-renewal to replenish the SC pool. HIF-1α pathway activation was found critical to maintain SC quiescence [99]. It was associated to an enhanced myogenic factor activation, an increased number of PAX7+ cells, a more rapid macrophage recruitment after a myotrauma [100] and could promote myogenesis by increasing *Myod* expression through the WNT pathway [101].

**Figure 4 ijms-22-07220-f004:**
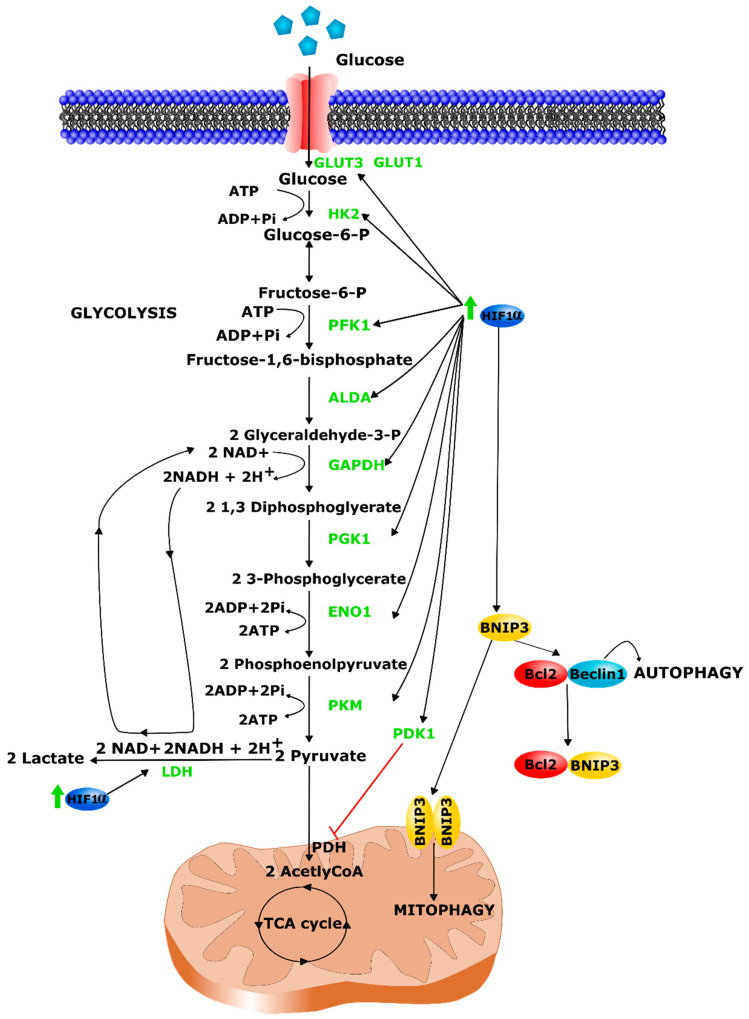
HIF-1α control on the glycolytic switch and mitophagy. HIF-1α promotes glycolytic metabolism through the induction of the expression of glycolytic transporters and enzymes (represented in green). Mechanistic aspects of the HIF-1α-BNIP3 induced mitophagy pathway.

**Figure 5 ijms-22-07220-f005:**
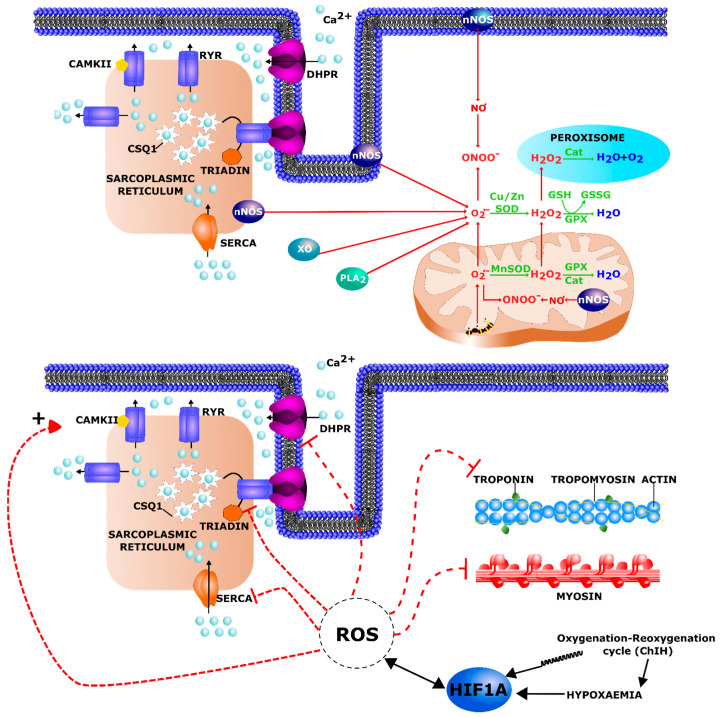
Upper part. Localization of the different ROS production sites and resulting ROS are linked by red arrows. Main enzymatic and non-enzymatic antioxidant defenses are represented in green. The main source of ROS is the mitochondria respiratory chain. Xanthine oxidase (XO) and neuronal NO synthase (nNOS) play also a large part in ROS production. Phospholipase A2 (PLA_2_) is activated by ROS and will be responsible for the hydrolysis of various products from the plasma membrane such as peroxidized fatty acids. The first antioxidant defenses are provided by superoxide dismutase (SOD) but the most important antioxidant is glutathione (GSH), a substrate of glutathione peroxidase (GPX) that neutralizes hydrogen peroxide by conversion into water. CSQ1 = Calsequestrin, the major calcium binding protein in the sarcoplasmic reticulum (SR). RyR = Ryanodine receptor, located in the SR membrane and responsible for the release of Ca^2+^ from the SR during excitation-contraction coupling. DHPR = dihydropyridine receptor, voltage-dependent Ca^2+^ channel located in T-tubule and also involved in excitation-contraction coupling. SERCA = sarcoplasmic reticulum Ca^2+^-ATPase allowing Ca^2+^ active transport from the cytoplasm to the lumen of the SR during muscle relaxation. Lower part. ROS sensitive molecular targets in skeletal muscle. ROS mainly modify muscle function by altering calcium concentration regulation and by oxidizing and consequently altering contractile muscle protein structure and function. Pro-oxidant environment can lead to the activation of CAMKII (Ca^2+^/calmodulin-dependent protein kinase II) which is known to cause RyR1 phosphorylation resulting in a leakiness of Ca^2+^ release from the SR.

**Figure 6 ijms-22-07220-f006:**
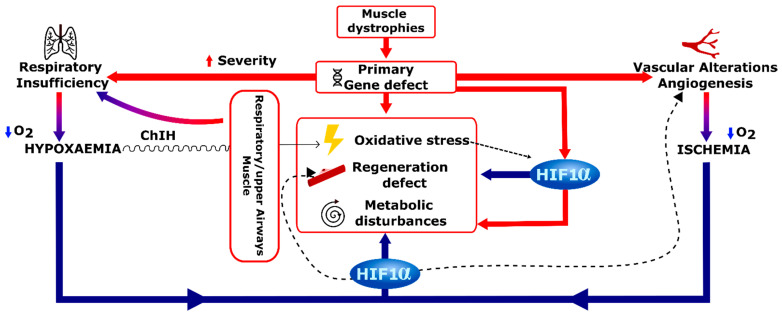
Overview of causes and consequences of hypoxia and HIF-1α activation in the context of MDs. The activation of hypoxic response pathways could emerge as a consequence of the primary genetic defect, or via the induction of hypoxia stress response pathways as a result of an indirect mechanisms e.g., respiratory insufficiency (inducing hypoxemia) and vascular alteration (causing ischemia).

**Table 1 ijms-22-07220-t001:** Main HIF-1α target genes and regulated pathways.

Regulated Pathways	HIF-1α Target Genes
Angiogenesis and erythropoiesis	VEGF and VEGF receptor FLT1, heme oxygenase 1, NOS 2 and 3, PDGF
Metabolism	ALDA, ALDOC, ENO1, GAPDH, HK1, HK2, LDH A, PFKL, PGK1, PKM, and TPI GLUT-1, 3, 4, PDK1
Proliferation and survival	Cyclin G2, TGFα and β3, IGF-2
Apoptosis	BNIP3/3L, P53
Myogenesis	WNT signaling

**Table 2 ijms-22-07220-t002:** Overview of respiratory involvement in muscular dystrophies. AD: autosomal dominant; AR: autosomal recessive; FVC: forced vital capacity; MEP: maximum expiratory pressure; MIP: maximum inspiratory pressure; PCF: peak cough flow; FEV1: forced expiratory volume in 1s; PEF: peak expiratory flow; TLC: total lung capacity; NIV: non-invasive ventilation; %VC: percentage of the predicted vital capacity; RV = residual volume; %p.v. = percentage of the predicted value; %c.v. = percentage of the control value. ↘ symbol means a decrease.

Muscular Dystrophy		Pathogenetic Factors		Clinical Characteristics		Respiratory Impairment	
		Inheritance	Affected Gene(s)	Muscle Distribution	Extra-Muscle Manifestations	Frequency	Type
Early onset							
Dystroglycanopathies (Walker–Warburg, Fukuyama muscular dystrophy, muscle–eye–brain disease)		AR	Dystroglycan and glycosy transferase enzymes genes	Primarily axial and limb muscles	Structural brain anomalies	Uncommon (12% in a study on 115 patients) [23]	Nocturnal hypoventilation and acute respiratory failure ↘ FVC (27 patients) [24]
Laminin-deficient muscular dystrophy		AR	LAMA2	Primarily upper limbs	Diffuse white matter hyperintensities on brain MRI and seizures	Frequent (30% of patients with complete laminin-a2 deficiency) [25]	Skeletal muscle weakness (including intercostal and accessory muscles), scoliosis and decreased chest wall compliance. Alveolar hypoventilation, mucus plugs with bronchial obstruction and atelectasis ↘ FVC (59 patients) [26]
SEPN1 myopathy (muscular dystrophy with rigid spine syndrome)		AR	SEPN1	Early rigidity of the spine and joint contractures of the ankle and elbow	Rigid spine, scoliosis	Frequent; early 81.7% requiring ventilation (132 patients) [27]	Diaphragmatic weakness ↘ FVC by 24 ± 7% (7 patients) [28]
Ullrich muscular dystrophy		AR	COL6A1, COL6A2, COL6A3	Primarily axial and limb muscles	Rigid spine, laxity of distal joints	Frequent; early	Diaphragmatic weakness ↘ %VC (40 patients) [29]
Childhood and Adult							
Duchenne muscular dystrophy		X-linked R	Dystrophin	Proximal lower limb and truncal weakness, followed by of upper limb and distal muscle weakness	Educational and psychosocial issue, scoliosis, cardiomyopathy and arrhythmias	Frequent	Vital capacity (% predicted) decreases linearly, due to inspiratory and expiratory muscle weakness. Obstructive sleep apnea and hypoventilation. Nocturnal desaturation correlated to the severity of scoliosis. ↘ FVC, FEV1 and PEF (115 subjects) [30]
Becker muscular dystrophy		X-linked R	Dystrophin	Same as DMD but with a milder phenotype	Less common than in DMD	Rare	Lung restriction sometimes occurs but less severe than in DMD
Emery–Dreifuss muscular dystrophy		Variable depending on type	EMD, FHL1, LMNA, SYNE1, SYNE2	Slowly and progressive humeroperoneal pattern	Cardiac conduction block, insulin resistance, rigid spine	Frequent; typically in adulthood	Restrictive pattern of respiratory impairment ↘ FVC to 60 and 45%p.v. (measured in 2 patients) [31]
Facioscapulohumeral dystrophy	FSHD1	AD	DUX4,	Facial, shoulder, scapular, arm progressive and asymmetric weakness	Retinal vasculopathy and symptomatic sensorineural hearing loss	First described as uncommon, 1–3% require NIV [32]. ↘ FVC in 38.3% and severely restrictive in 14.9% [33].	Expiratory and diaphragmatic muscle weakness and obstructive sleep apnea ↘ mean FVC to 69%p.v. in non-mild disease (40.2%p.v. in early onset), minimum 33%p.v. (adult) and 11%p.v. (early onset) [33] ↘ MIP (69%c.v.), MEP (53%c.v.) and PCF (60%c.v.) [34]
	FSHD2		Digenic: DUX4 + either SCHMD1, DNMT3B or LRIF1				
Limb girdle muscular dystrophies		AR more frequent than AD	Sarcoglycan, Dystroglycan, Telethonin, Titin, etc.	Variable but mostly proximal weakness	Cardiomoypathy (common in sarcoglycan deficiency and dystroglycano pathy)	Common in sarcoglycan	Respiratory insufficiency due to diaphragmatic weakness Restrictive pulmonary syndrome indicated by TLC < 80%p.v. (13/38 patients) [35] FVC below 40%p.v. (20/38 patients) ↘ PEF (38 patient study)
Myotonic dystrophy		AD	DMPK, CNBP	Distal slowly progressive weakness	Cardiac dysrhythmia, particularly heart block	Common	Sleep apnea syndrome and excessive daytime sleepiness ↘ MEP (21 patient study) [36] ↘ FVC, VC, TLC, RV, FEV1 [37]

**Table 3 ijms-22-07220-t003:** Effect of hypoxia on myogenesis in vitro.

Experiments In Vitro				
Species	Cell Type	Way of HIF-1α Stabilization	Effect on Myogenesis	Ref.
Mouse	C2C12	Hypoxia at 5% O_2_	No effect	[106]
		Hypoxia at 2% O_2_	↘ differentiation with ↘ *Myod* and *Myog* expression	
		Hypoxia at 0.5% O_2_		
		Hypoxia at 0.01% O_2_		
Mouse	C2C12	Hypoxia at 0.5% O_2_	↘ differentiation with ↘ *Myod*, *Myog* and *Mhc* expression	[107]
Mouse	C2C12	Hypoxia at 1% O_2_	↘ differentiation with ↘ *Mhc* expression dependent on notch signaling	[108]
Mouse	Primary myoblast	Hypoxia at 1% O_2_	↘ differentiation through p53-dependent induction of *Bhlhe40*	[109]
Mouse	C2C12	Cobalt chloride	↘ differentiation with ↘ myoblast proliferation, ↘ *Myog* expression	[112]
Mouse	C2C12	Hypoxia at 5% O_2_	↘ differentiation with ↘ *Myod*, *Myog* and *Mhc* expression	[113]
		Hypoxia at 10% O_2_	↗ differentiation with hypertrophy and ↗ *Myog* and *Mhc* expression	
		Hypoxia at 15% O_2_		
Mouse	C2C12	Hypoxia at 1% O_2_	↘ differentiation with ↘ *Myod, Myf5, Myog* and *Mhc* expression	[111]
Rat	L6	Hypoxia at 1% O_2_	↘ differentiation with ↘ myoblast proliferation and ↘ myogenic index	[110]
Rat	L6E9	Hypoxia at 1% O_2_	↘ differentiation with ↘ *Myod, Myf5, Myog* and *Mhc* expression	[111]
Human	Primary myoblasts	Hypoxia at 1% O_2_	↘ differentiation with ↘ myoblast proliferation and ↘ myogenic index	[110]
Bovine	SCs	Hypoxia at 1% O_2_	↗ differentiation with ↗ SC proliferation and ↗ *Myod*, *Myog* and *Mhc* expression	[115]

↘ symbol means a decrease. ↗ symbol means an increase.

## Data Availability

Not applicable.

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
