# Peer review of "Hypoxia and Hypoxia-Inducible Factor Signaling in Muscular Dystrophies: Cause and Consequences"

_ijms, 2021, doi:10.3390/ijms22137220_

Round 1

Reviewer 1 Report

This is a potentially interesting review that provides insights into the role of hypoxia and the HIF-pathway in muscular dystrophies. Unfortunately, the text reads in sections as a pure collection of information without putting them into an easily comprehensible context for the reader.

My specific comments are:

major comments:

  • Table 1: Please state that the table is far from being complete.
    • EPO is more likely to be a HIF-2 target gene.
  • Figure legend 1 and line 64: it is not described why PHD are inhibited in hypoxia (PHD enzymes are oxygen dependent). Please clarify.
  • The work of William G. Kaelin Jr., Sir Peter J. Ratcliffe and Gregg L. Semenza should be cited when the HIF pathway is described (Nobel price in medicine 2019 for the discoveries of how cells sense and adapt to oxygen availability)
  • e.g. lines 88, 91, 94: “HIF-1a activation” should be “HIF pathway activation” as HIF-1a is rather stabilized in hypoxia.
  • Fig 3: How is the HIF-pathway involved in the different steps of myogenic differentiation? It would be nice if this information could be added to the figure.
  • Paragraphs starting at line 364 and line 372 need some work.
    • The outline in these paragraphs is confusing to me. Perhaps a table summarizing the way the HIF-pathway is activated, cell type, effect on differentiation could help the reader to get an overview.
    • For chemical hypoxia, it should be pointed out that very likely HIF-independent effects are also provoked.
    • line 364; 446“skeletal muscle expresses HIF1a…” Do the authors mean: In skeletal muscle HIF-1a is stabilized under normoxic conditions…?
    • pO2 is the partial pressure of oxygen. The unit of a pressure is not %. (e.g. lines 378, 386, 387)
    • lines 385-387:It is not clear to me which point the authors want to make. C2C21 cells are a cell line and has like all cell lines that are kept in the usual cell culture incubators reset its normoxic setpoint to 21% O2.
    • Please double check if “physioxia is 4%” in muscle (line 387).
  • line 462: “Accordingly, Phd2 KO mice had better endurance performance …”The endurance performance can´t be discussed without mentioning that EPO is induced. The fiber type switch is most likely not the only reason for the better endurance of Phd2 ko mice.
  • Figure 4 and paragraph starting at line 586: What happens to HIF-pathway induced enzymes in MDS?
  • paragraph starting at line 593: Can the HIF-pathway affect Ca2+ homeostasis?
  • Fig. 5: Is CSQ1 mentioned anywhere in the text?
  • line 761; HIF as potential therapeutic target for muscle disorders: What is needed? HIF pathway activation or inhibition?

minor comments:

  • check spelling of HIF-1a
  • Figure 1: Hydroxylation is shown by adding OH groups to HIF. The lines should be drawn to the “O” instead of the “H”.
  • Sentence line 159 -161: “The compliance of the respiratory system described the importance of the change in lung volume for a given pressure generated…”. Could the authors please clarify what is meant by this sentence. The compliance of a system is defined as the change in volume that occurs per unit change in the pressure of the system.
  • Sentence line 163 -165: “ In addition….” Reference is missing
  • lines 181-182: “Altogether, lung and chest…”. Could the authors please check if this sentence is correct

Reviewer 2 Report

This is a comprehensive review by Tassin and colleagues, where they have discussed molecular components and physiological relevance of Hypoxia-inducible Factor signaling in muscular dystrophies. The article is well organized, well written and discusses most of the recent literature. 

I only suggest the Authors to include a novel paragraph on the pharmacological regulators of HIF1a in skeletal muscle disorders.

Round 2

Reviewer 1 Report

The revised manuscript has been improved. I believe the manuscript is now acceptable for publication.